# Intrinsic and extrinsic motivations governing prey choice by hunters in a post-war African forest-savannah macromosaic

**Franciany Braga-Pereira**[ORCID][1,2,3] *, **Carlos A. Peres**[4,5], **Rômulo Romeu da Nóbrega Alves**[1,6], **Carmén Van-Dúnem Santos**[3]

**1** Department of Ecology and Systematics, Universidade Federal da Paraíba, João Pessoa, PB, Brazil, **2** Rede de Pesquisa para Estudos sobre Diversidade, Conservação e Uso da Fauna na Amazônia (RedeFauna), Manaus, Amazonas–Brasil, **3** Agostinho Neto University, FCUAN, Luanda, Angola, **4** Instituto Juruá, Aleixo, Manaus, AM, Brazil, **5** School of Environmental Sciences, University of East Anglia, Norwich, United Kingdom, **6** Department of Biology, Universidade Estadual da Paraíba, Campina Grande, Brazil

* franbraga83@yahoo.com.br

**Data Availability Statement:** Data cannot be shared publicly because it contains potentially identifying details of the poachers. Data are available from the Research Ethics Committee

## Abstract

Overhunting typically increases during and after armed conflicts, and may lead to regional-scale defaunation. The mitigation of hunting impacts is complex because, among other reasons, several intrinsic and extrinsic motivations underpin the elevated deployment of hunting practices. Here we present the first study focusing on these motivations in a post-war zone. Following persistently heavy hunting pressure during the 27-year Angolan civil war, the offtake of small to medium-bodied species has increased recently as a result of large mammal depletion. However, prey choice associated with different motivations varied in terms of species trophic level and body size. While most residents hunted large-bodied species to maximize revenues from wildlife trade, many low-trophic level smaller species were harvested to meet local subsistence demands because they were more palatable and could be captured using artisanal traps near hunters' households. Mainly low-trophic level species were killed in retaliation for crop-raiding or livestock depredation. Considering all game species sampled in this study, 96% were captured to attend two or more motivations. In addition, hunting associated with different motivations was partitioned in terms of age and gender, with prey acquisition for the wildlife trade primarily carried out by adult men, while hunting to meet local subsistence needs and inhibit human-wildlife conflicts were carried out by adult men and women, children and even the elderly. In natural savannah areas lacking fish as a source of protein, a higher number of species was selected to supply both the meat trade and subsistence, while more species in forest areas were targeted for trade in animal body parts and conflict retaliation. Finally, local commerce in bushmeat and other body parts accrued higher domestic revenues compared to any alternative sources of direct and indirect income. However, these financial benefits were at best modest, largely unsustainable in terms of prey population collapses, and generated high long-term costs for the local to regional scale economy and native biodiversity.

(CEP) / Health Sciences Center (CCS) of the Federal University of Paraiba (UFPB) (contact via comitedeetica@ccs.ufpb.br) for researchers who meet the criteria for access to confidential data.

**Funding:** FB-P was supported by the Faculdade de Ciências-Universidade de Agostinho Neto (Angola), by The Rufford Foundation (ID 20153-1) and by an MSc scholarship granted by the Brazilian Ministry of Education (CAPES, 20151025938). CVDS was supported by Faculdade de Ciências-Universidade de Agostinho Neto (Angola). RRNA was supported by a productivity research grant of National Council for Scientific and Technological Development (CNPq). Universidade de Agostinho Neto: https://uan.ao/ The Rufford Foundation: https://www.rufford.org/projects/franciany-gabriella-braga-pereira/hunting-and-use-of-wild-mammals-by-human-communities-in-a-conservation-area-in-angola-the-influence-of-environment-on-hunterss-niche-breadth/ CAPES: https://www.gov.br/capes/pt-br CNPq:https://www.gov.br/cnpq/pt-br The funders had no role in study design, data collection and analysis, decision to publish, or preparation of the manuscript.

**Competing interests:** The authors have declared that no competing interests exist.

# 1 Introduction

Hunting of wild terrestrial vertebrates, as one of the oldest activities pursued by both archaic and modern humans, has been undoubtedly critical in human ecology and evolution because hunting represents a form of defence against wild animals and procurement of food, clothing and therapeutic products [1–3]. However, hunting practices have resulted in large-scale defaunation, and even subsistence hunting to meet local needs represents one of the primary drivers of local extinctions of wildlife populations [4–6]. In many tropical regions, hunting exerts become a more severe threat to native wildlife than primary habitat loss via the conversion of natural vegetation [7,8]. Moreover, the potential ecological effects of overhunting are not only on populations of game species, but also on the overall structure and dynamics of natural ecosystems [5,9], and overhunting places local livelihoods and food security at risk [10]. Large-bodied mammals are often the most frequently selected prey species for the bushmeat trade due to their larger meat yields and greater returns on energy invested in hunting [11,12], in addition to being more intrinsically detectable for game hunters in open landscapes [13,14]. However, these mammal species generally exhibit lower fecundity rates, larger home ranges and lower population densities, which result in higher vulnerability to population depletion and local extirpation [15–17]. Species-specific hunter selectivity is often also both size- the sex-biased, so that the impact of hunting tends to be higher on the survival of sexually mature males, particularly in size-dimorphic polygynous species [18]. However, selective hunter-induced mortality can also be markedly sex-biased in weakly dimorphic species depending on the hunter preferences and hunting regulations, if any [19,20].

Overhunting has increased in the last three decades partly due to widespread road expansion, more affordable transportation and hunting technology, population growth, higher market demand, and the use of more efficient hunting methods [20–22]. In addition, all of these aggravating factors tend to be correlated with each other. For example, the contemporary use of shotguns in some traditional communities in Cameroon is related to the emergence of more specialized and highly efficient hunters with clear market orientation [23]. A critical element of effective conservation is therefore a proper understanding of the range of motivations propelling hunting behaviour in humans [24].

A comprehensive list of typologies of illegal hunting motivations could include household consumption, commercial gain, recreational and trophy killings, protection of self and property, poaching as a traditional right, gamesmanship, disagreements with specific regulations and subversive poaching [25]. Moreover, we can essentially distinguish hunting motivations into either intrinsic, when the demand for game kills arises from *in situ* concerns in which the hunters themselves are embedded, or extrinsic, which arises from a motivation outside the community and often accounts for the largest fraction of the animal biomass offtake [26,27]. For example, the perceived risk of livestock predation by wolves in Spain and jaguars in Brazil are key motivations for the systematic poisoning of these animals [28]. In addition, wild meat (including terrestrial and aquatic vertebrates) represents the main source of protein in the daily diet of traditional populations worldwide [29], and is critical for the well-being and nutrition of local communities [30]. Hunting intended for self-consumption also includes animals selected for therapeutic purposes [31], amounting to intrinsic motivations to hunt. On the other hand, commercial offtake of elephants and rhinos is primarily motivated by the exceptional market value of ivory and horns, respectively. Moreover, the illegal trade in bushmeat and other body-parts has led to many social and economic problems as local prices of commercially valuable species and their derivatives continue to rise, which in turn generates ever greater incentives for this illegal practice [32,33].

Individual-based motivations can become even more complex at sites experiencing episodes of socio-political upheaval, which can increase demand in meat and other body parts

and effectively lift restrictions on game hunting. For example, during civil wars in low-governance countries (outside of land mine zones) hunters become more encouraged to hunt due to sociopolitical changes induced by the uprise of conflicts (e.g. suspension of anti-poaching patrols, increased access to automatic weapons, and growing demand in urban markets) [14,34]. Increased market demand can be fueled by diverse sets of consumers, including rural war refugees, urban elites, and the international gourmet meat market. In addition, high-value animal by-products from the illegal wildlife trade (such as elephant ivory and rhino horns) are often exchanged to finance militias. During the post-war period, however, large mammal populations will continue to decline unless there is a concerted cooperative effort to protect natural ecosystems and their wildlife [14,35]. Regulation of high-value trade in threatened species also requires coordinated efforts to manage procurement.

Few studies have focused on these intrinsic or extrinsic motivations underlying hunting and/or the mechanistic motivations underpinning game pursuit and choice, and ultimately why some mammal populations are more impacted than others. In addition, until recently, aside from anecdotal reports, there are few published data on hunting in Angola during and post-civil war, although hunting is a common practice in the country [36]. Literature increasingly suggests the use of interviews as a type of method based on local ecological knowledge (LEK) to quickly understand human perceptions and motivations toward wildlife. These methodologies are accurate, cost and time effective and can rapidly inventory large areas [37–39].

Here, we examine the patterns of hunter motivation in a post-war zone in relation to species body mass and trophic position through interviews, including capture rates of individuals in different sex and age classes. Considering that conservation problems are inextricably linked to socioeconomic context, we also examine the degree to which the illegal wildlife trade has been more profitable in the short term than alternative livelihood activities pursued by local residents. To do so, we conducted interviews using a robust sample size of local hunters within the two most important protected areas of Angola, Southwest Africa. Finally, we show that different forms of motivation bias hunting selectivity in terms of species body mass, trophic level, and availability and further discuss the economic drivers behind hunting practices.

## 2 Material and methods

### 2.1 Study landscape

This study was carried out at Quiçama National Park and Quiçama Game Reserve, which span a total area of 960,000 ha (9˚09'–10˚23'S and 13˚09'–14˚08'E) and represent the most important conservation areas of Angola [40]. We selected eight human settlements distributed throughout this area (Fig 1). A government census showed that there were 25,086 residents in the Quiçama municipal county in 2014. Although this local population is not legally permitted to occupy these two protected areas, they have done so for generations, which creates tensions between residents and the reserve management. The semi-subsistence economy of the households in the area is sustained by hunting, small-scale fisheries, slash-and-burn cassava agriculture, and non-timber forest products. In early 2000, a rehabilitation project (Operation Noah's Ark) conducted a species translocation program to the Special Conservation Area, but unfortunately many of these species were exotic to the park [41]. These introduced and reintroduced alien species are not considered in this study.

### 2.2 Ethics approval

This research was approved by the i) National Health Council (Resolution 466/12), through the Research Ethics Committee of the Universidade Federal da Paraíba (license number

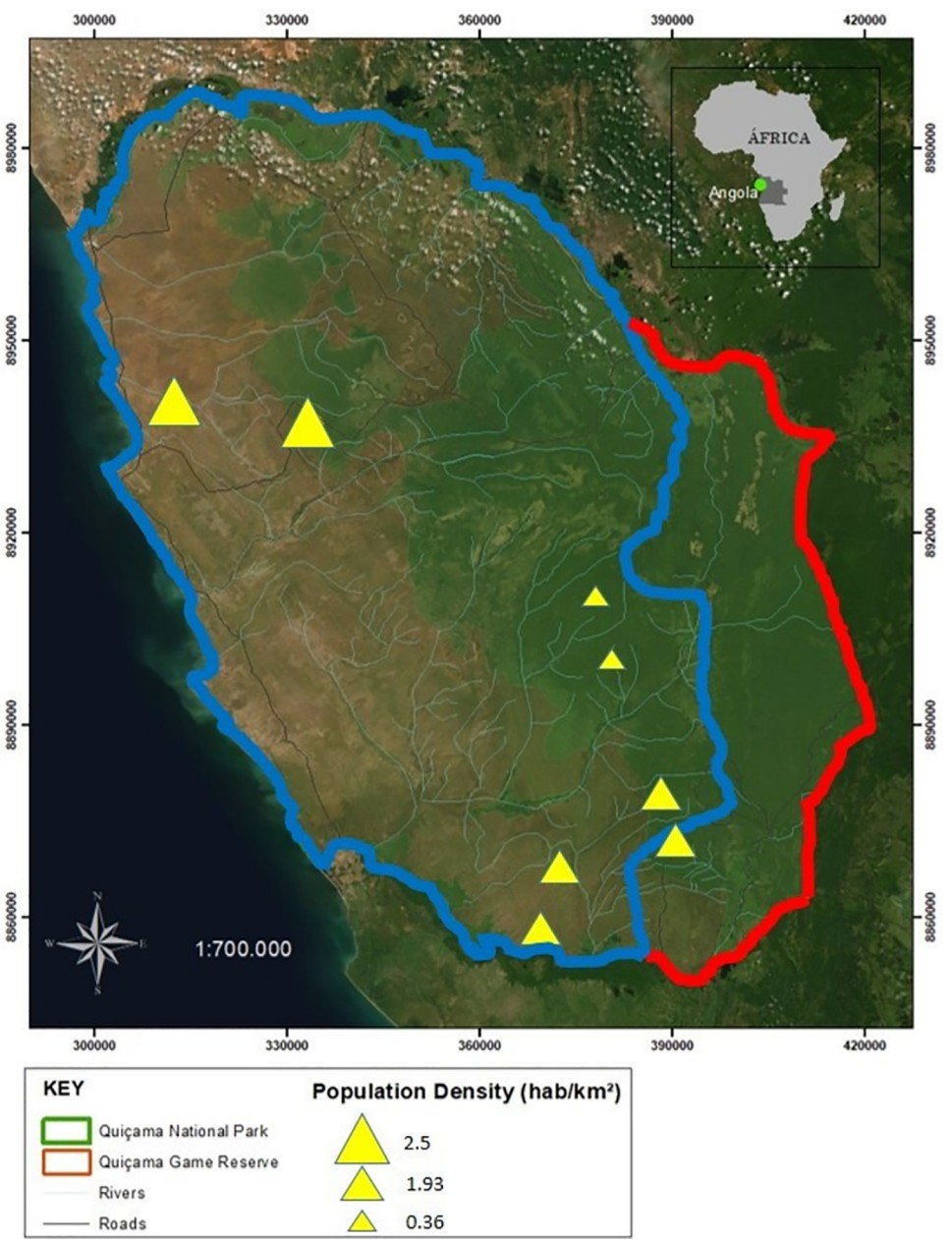

**Fig 1. Location of Quiçama National Park (demarcated by the blue line) and Quiçama Game Reserve (red line) in Southwest Africa.** Solid yellow triangles indicate the surveyed townships, which encompassed more than one human settlement. Triangles sizes are proportional to the population size of the townships. Green and brownish background areas represent forest and savannah environments, respectively. Map generated using ArcGIS 10.3.1; Datum: WGS84. Elaborated by Ana Caroline Imbelloni; Franciany Braga-Pereira in August/2021.

59846816.3.0000.5188); ii) Environmental Ministry of Angola (148INBAC.MINAMB/2016); iii) municipal administration of Quiçama (license number 017/GAB.ADM.MQ/2017); iv) leaderships of all sampled villages. An informed consent form was given to the interviewee, stating the purpose of the interview and ensuring the interviewee's anonymity, so each interview could not begin until after this form had been signed.

## 2.3 Data collection

We used Local Ecological Knowledge (LEK) to obtain information on hunting and wildlife use. The first contact between FBP and community dwellers occurred in 2014 to delineate the study area and present proposals to local villagers prior to deployment of this study. This facilitated access to the study area and a relationship of trust with local informants, including dedicated and semi-professional hunters. Data for this study were obtained from January 2014 to January 2015 and from January to April 2017. Translators were not required because both the interviewer and local informants spoke Portuguese. Interviews were conducted with 115 experienced local hunters (112 men and 3 women), who ranged in age from 19 to 80 years old. These hunters were selected using a snowball sampling technique [42], and on average we interviewed 14 informants per settlement. In addition to engaging in hunting activities, interviewees also occasionally worked as small-scale farmers, fisherfolk, teachers and community leaders.

Field data were collected through semi-structured individual interviews [43] with the help of a visual stimulus checklist using colour photos of mammals living within the research area. For each game species reported as killed by the interviewed hunter, he/she was asked to provide information on the (i) the date of the most recent mammal specimen killing; (ii) sex and age class of the most commonly hunted individuals; (iii) motivation behind the pursuit each wild mammals; and (iv) the revenue acquisition from the sale of the animal when traded including the trade value (per kg) as: a) live whole animal, b) whole dead animal (e.g. fresh meat), c) processed as dried meat, d) processed skin, and e) ivory. We were therefore able to quantify the total aggregate value of each prey item. Each hunter was also asked about sources and amounts of income acquired from wage labour and other activities he or she may have performed, including farming, teaching and community leadership (see the complete questionnaire in S1 File).

## 2.4 Data compilation

As all questionnaire answers were open-ended, in order to facilitate data analysis different answers to the question "what is the motivation behind the pursuit [this] species?" were classified into common themes following Braun and Clarke [44]. Coded responses were summarized, thereby resulting in five hunting motivation categories: meat trade, pet trade, other trade in body parts, subsistence needs, and conflict retaliation. This method of open-ended questioning provided the flexibility to explore, whenever necessary, different lateral topics of relevance while providing rapid anthropological assessments valid for wildlife research. Finally, we organized a binary data frame structured as species and interviewees in rows and hunting motivations in columns, in which the absence or presence of a specific motivation associated with a specific species, according to each interviewee, was attributed the values 0 or 1, respectively.

We used the PanTHERIA database [45] to obtain information on body mass and trophic level of each species. Based on specific morpho-ecological traits, all mammal species were assigned to seven dietary functional groups. We initially ranked the energetic stratum of the modal dietary pattern of each species as follows: (1) folivore < (2) frugivore < (3) granivore < (4) insectivore < (5) myrmecophage < (6) mesocarnivore < (7) hypercarnivore. We then weighted the proportion of each major dietary mode of any given species (sourced from [46,47] given these food energy levels (e.g. if the grazer/frugivore *Tragelaphus oryx* consumes 70% foliage and 30% fruit, its mean trophic energy level would be 1.3 (= $(0.7 \times 1) + (0.3 \times 2)$). We used the PanTHERIA and AnAge databases [48] to obtain information on species litter size and number of litters per year. Per capita annual fecundity (defined as female young per

adult female per year) were therefore calculated as (litter size × number of litters per year)/2 (i.e. assuming a 50:50 birth sex ratio) [49]. The monetary value of each animal is expressed as the mean transaction value per individual reported for each species by all independent interviewees. For details of English and scientific species name, and species body mass annual fecundity rate, trophic level and monetary value acquired per animal see S1 Table.

We assigned the 8 sampled communities into four study sites according to type of landscape structure and geographic position. We therefore sampled two communities in savannah areas of northern Quiçama; two in savannah areas of southern Quiçama; two in central forest areas; and two communities in southern forest areas. Some differences between these sites include: the northern savannah communities are near (~20 km) a semi-urbanized area, and influenced by a nearby paved road. The central/southern forest sites had access to the highest large mammal abundance in Quiçama and access roads were in poor condition. The southern forest sites are near lakes and small streams, so the local population could also meet their protein needs from fish. The southern savannah site, although far from an urban center, had access to a road in good condition, thereby facilitation trade. There were some lakes and small rivers at this site, which could also supply some animal protein from fish.

## 2.4 Data analysis

**Species traits associated to each hunting motivation.** To examine the effects of body mass and trophic level on the number of hunters reporting each hunting motivation for killing any given species we performed generalized linear mixed models (GLMMs). We performed five different models associated with each hunting motivation. We considered i) the hunting motivation as a response variable; ii) species body mass and trophic level as fixed effects predictor variables; and iii) interviewee as a random factor. Given that this is a binary response (0 or 1), we used binomial distribution. There was no observed collinearity (p >0.05) among predictor variables. We used residual checks to verify whether our models were, in principle, suitable or otherwise. We used the Akaike information criterion to select models of interest. The model with the lowest AIC was retained, and the remaining competing models were ordered according to their Akaike differences (ΔAIC) with respect to the best model (lowest AIC) [50,51]. To test whether different hunting motivations, and species body size and trophic level were related to each other, we performed multiple factor analysis (MFA) [52]. Compared to the frequently used principal component analysis (PCA), MFA takes into account the fact that data are structured into groups (here different species) to balance the importance of each group in the analysis. Species traits then become more important when any given species is repeatedly cited by several interviewees.

**Local revenues from wildlife offtake.** We performed linear models (LM) to examine the strength and direction of linear relationships between prey body size and fecundity rate; body size and local revenues accrued from sales of each species; and fecundity rate and local revenues. Given that only one predictor variable was explored in each LM, we used frequentist statistics to assess variable effects, including F-values, degrees of freedom, p-values and adjusted $R^2$-values.

**Hunter demography associated with hunting motivations.** To examine the effects of age, gender and natal community on the number of hunters reporting killing motivations for different species, we performed MFAs. So, we evaluated the degree to which hunting motivations were the same across all these demographics variables.

All analyses were performed in R ver. 3.5.3 (R Core Team 2019). GLMMs and GLMs were based on the *lme4* [53] package and MFAs were based on the *FactoMineR* [52] package.

# 3 Results

## 3.1 Prevalence and history of kills and game choice

Species reported as captured by all interviewed hunters were Bushbuck Kewel and Bushpig. Blue Monkey and Blue Duiker were also killed by 94.7% and 93.0% of all interviewed hunters, respectively. Species killed by a small number of hunters included Hippopotamus (16.5%) and Elephant (10.4%). Most of the recent capture records for small and medium-sized species had occurred within the previous 10 years, with few exceptions such as Serval, Wild dog and Hyena. These former species followed the pattern of large mammals, for which the most recent captures occurred within 20 to 50 years prior to interviews (Fig 2). Adult animals were far more frequently selected, as reported by 89.6% of all informants, with the remainder indicating that animals were either selected regardless of age class (8.0%) or that young animals were more frequently selected (2.4%). In relation to sex classes, most informants (54.9%) reported that there were no differences in the frequency with which either males or females were selected, although 39.5% indicated that males were mainly targeted, with only 5.6% of all informants indicating that females were more frequently selected.

## 3.2 Motivations driving prey acquisition

From those hunters reporting intentionally capturing each species, the main intrinsic motivations were subsistence meat acquisition and retaliation against wildlife posing perceived threats; while extrinsic motivations were bushmeat trade, and trade in animal body-parts and

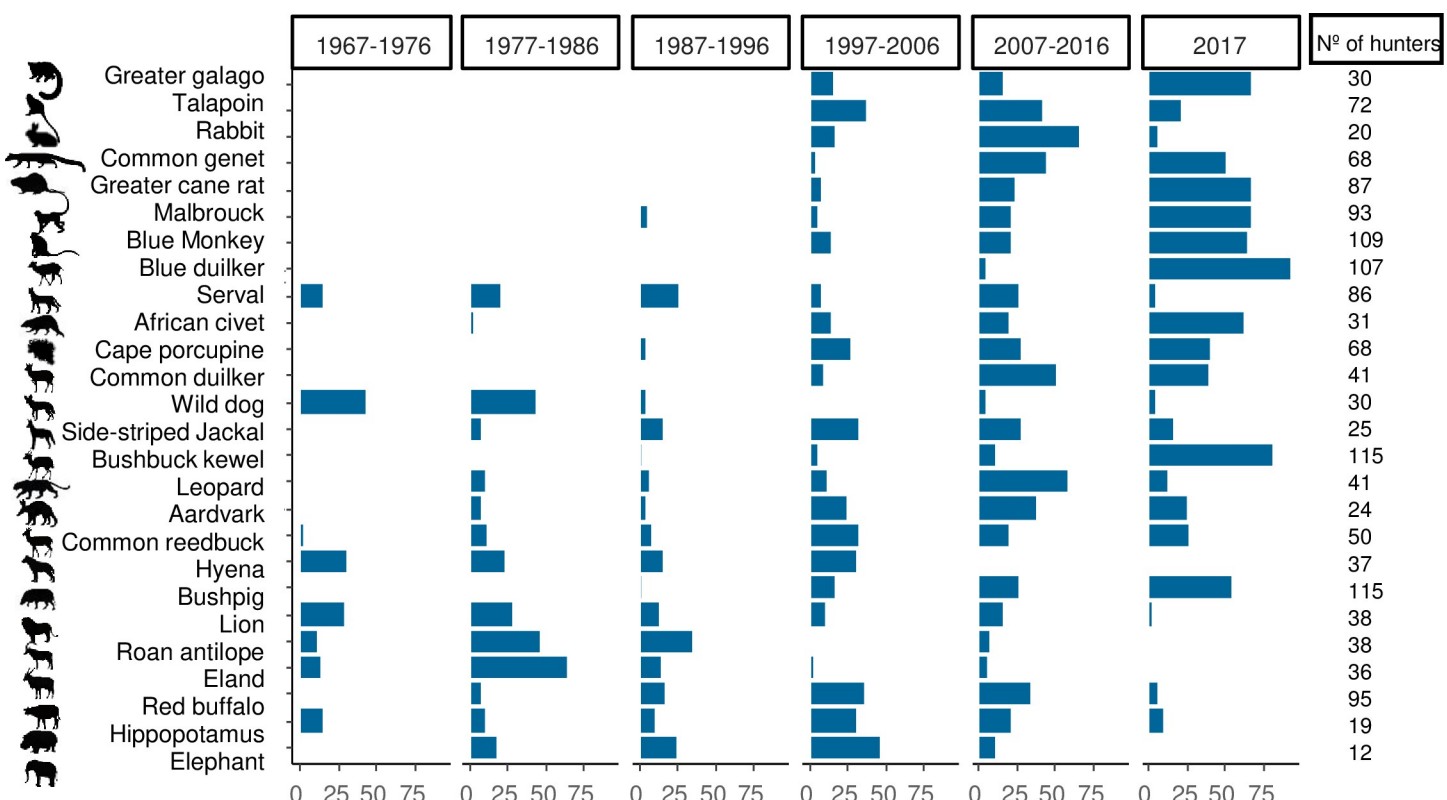

**Fig 2. Records of the most recent year in which each interviewee had killed a mammal species.** Data are presented in 10-year increments for historical (1967–2016) and modern periods (2017). Response rates were calculate from the number of hunters reporting killing each species. Species are denoted by their English names and ordered top to bottom from the smallest to the largest species.

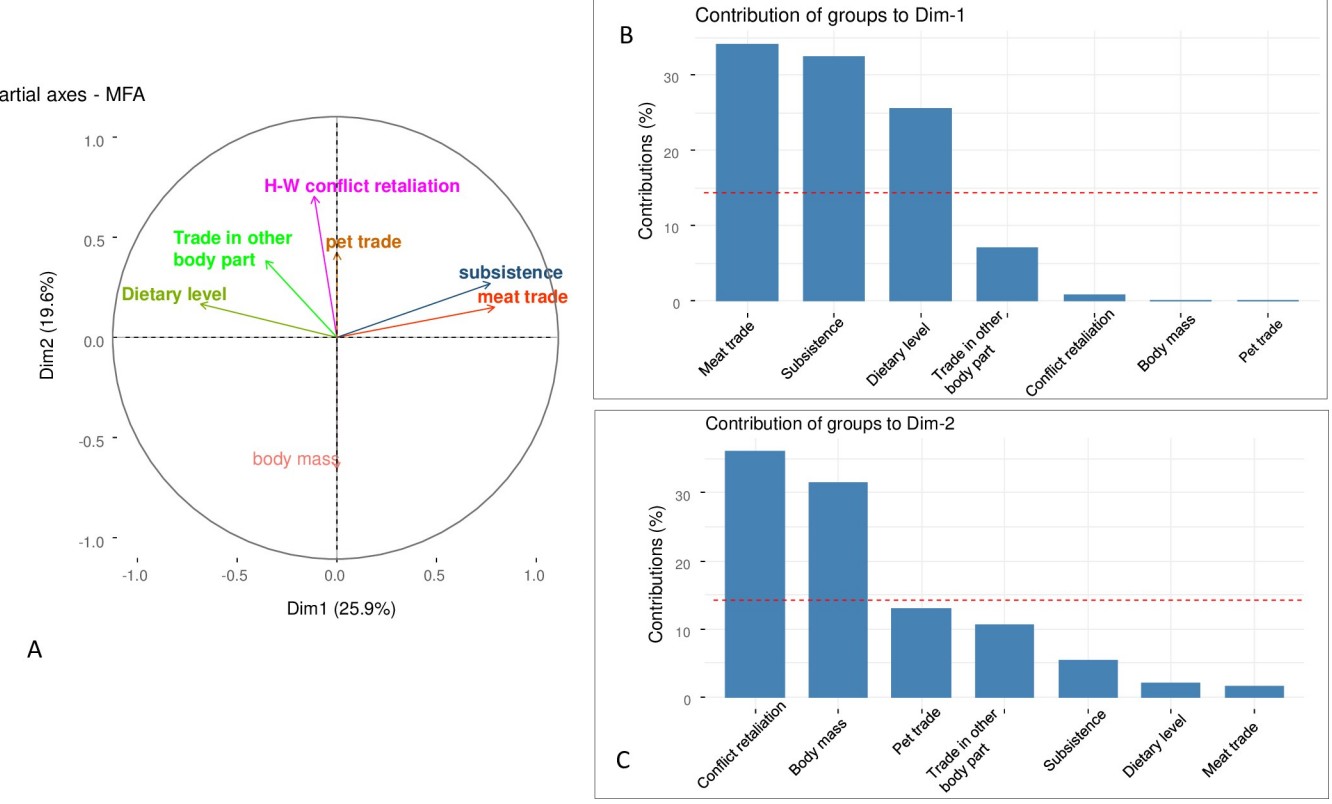

**Fig 3.** Species compositional similarity in terms of hunting motivations obtained from (a) a multiple factor analysis (MFA). Contributions of each hunting motivation and species trait for (b) the first ($Dim_1$) and (c) the second dimension ($Dim_2$). The red dotted line indicates the percentage that would be obtained if all factors contributed equally to the overall variance.

pets. With the exception of Greater Galago, all other game species were captured to attend two or more motivations and seven species to attend for out of the five motivations (Common Duiker, Bushpig, Blue Monkey, Cape Porcupine, Malbrouck, Greater Cane Rat and Talapoin) (S1 Fig). Some hunters also used some animal body parts for zootherapeutical purposes, but this was rarely the primary motivation for killing any given species. All respondents reported killing at least one species for food, retaliation and sales of meat. However, trade in body parts was restricted to only 40% of all informants.

Species hunted to supply both subsistence needs and the meat trade were trophically similar and primarily herbivores at low dietary energy levels. The first dimension ($Dim_1$) of the MFA accounted for 25.9% of the variability in hunting motivation across different species (Fig 3A). The meat trade, subsistence consumption and dietary level contributed 34%, 32% and 25% of the variance explained by $Dim_1$, respectively, whereas procurement for other body parts, the pet trade, retaliation against "problem" species and prey body size combined contributed with only 10% (Fig 3B). The second MFA dimension ($Dim_2$) explained 19.6% of the variance with retaliation and body size explaining most of the variance (37% and 32%, respectively), whereas other factors were relatively unimportant (pet trade 14%, trade in body-parts 10%, subsistence needs 5%, dietary level 2% and the meat trade 2%; Fig 3C).

Local meat consumption was associated with the killing of species at low trophic levels (p< 0.001), yet harvested prey spanned the entire spectrum of body mass, from Cane Rats (4 kg) to Bushpigs (69 kg). Killings justified as retaliation against human-wildlife conflicts also involved mainly species at low trophic levels (p< 0.001), across all body mass spectrum but mainly

smaller-bodied species. The reasons of retaliation against HW-C included crop-raiding (e.g. primates such as Blue Monkeys and Malbrouck Monkeys) and predation on domestic livestock, fish or wild animals of human interest (e.g. felid depredation on chickens). For the bushmeat trade, species at low trophic levels were cited as significantly positively selected (p< 0.001), particularly large-bodied species. Kills to fuel the trade in animal body parts included mainly felids (p< 0.01) of varying sizes and intentional captures of primates were also motivated by the pet trade (Fig 4; Table 1). Animal products were often sold to intermediaries who, in turn, traded at higher profits outside local communities. Although informants were unsure about reporting the ultimate motivation behind felid skeletal parts traded, they believed those would be used largely for ornamental or medicinal purposes either within Angola or other countries. In addition to Carnivores, other prey species provided body parts that were secondarily used for medicinal purposes by local communities. However, Honey Badgers (*Mellivora capensis*) were reportedly killed primarily for therapeutic purposes (Table 2).

### 3.3 Local revenues from wildlife offtake vs other activities

Large-bodied species yielded the highest household revenues obtained from sales of single prey items ($F_{1,17}$ = 299.2; p< 0.001; $R^2_{adj}$ = 0.94). To make matters worse, these large mammals often exhibit low fecundity rates (Fig 5). The five most valuable prey species fetched an average market price per individual carcass of US$ 1359 (range = US$ 638–3117 per individual). These species, however, were becoming scarce and had not been recently captured by local hunters (Fig 2).

Compared to revenues provided by medium and large-bodied species, local revenues accrued from any alternative wage labour option were very modest and well below the poverty line for Angola, where monthly income are US$98, US$43, and US$33 for teachers, farmers, and community leadership roles, respectively. In comparison, the value accrued from a single kill exceeding 50 kg was considerably greater. For example, annual revenues from gazelle kills (*T. scriptus)* alone were twice that from teaching, and the sale value of a single leopard skin was nearly four-fold higher than the monthly wages of a primary school teacher, who earned a higher income than either a farmer or community leader.

**Hunter traits and hunting motivations.** In relation to the sites (regarding landscape structure and geographic position), we found a higher number of kill reports in northern savannah areas were due to subsistence and the meat trade than in either southern savannah and forest sites. Forest sites showed a higher number of prey offtake reports induced by retaliation killings and trade in other body parts compared to savannah sites. The MFA analysis showed a high overlap of species selected for the meat trade and subsistence in northern savannahs, and a similar set of species killed due to the body-part trade or conflict retaliation across all forest sites (S2 Fig). Although middle-aged male hunters more frequently reported mammal kills, sex and age classes of informants had a low contribution in explaining MFA dimensions 1 and 2 (S3 Fig). Hunting for the wildlife trade was primarily done by adult men, whereas kills attributed to local crop-raiding or livestock depredation and local subsistence, were also carried out by adult women, children and the elderly.

## 4 Discussion

In the early stage of the Angolan civil war, target mammal species harvested at the Quiçama National Park region of Angola consisted primarily of medium- to large-bodied species. This was consistent with the set of game species sold in the markets in Luanda (capital of Angola), which were commercially valuable as either bushmeat or other body parts. However, as large-bodied mammals became increasingly overhunted and their populations commensurately

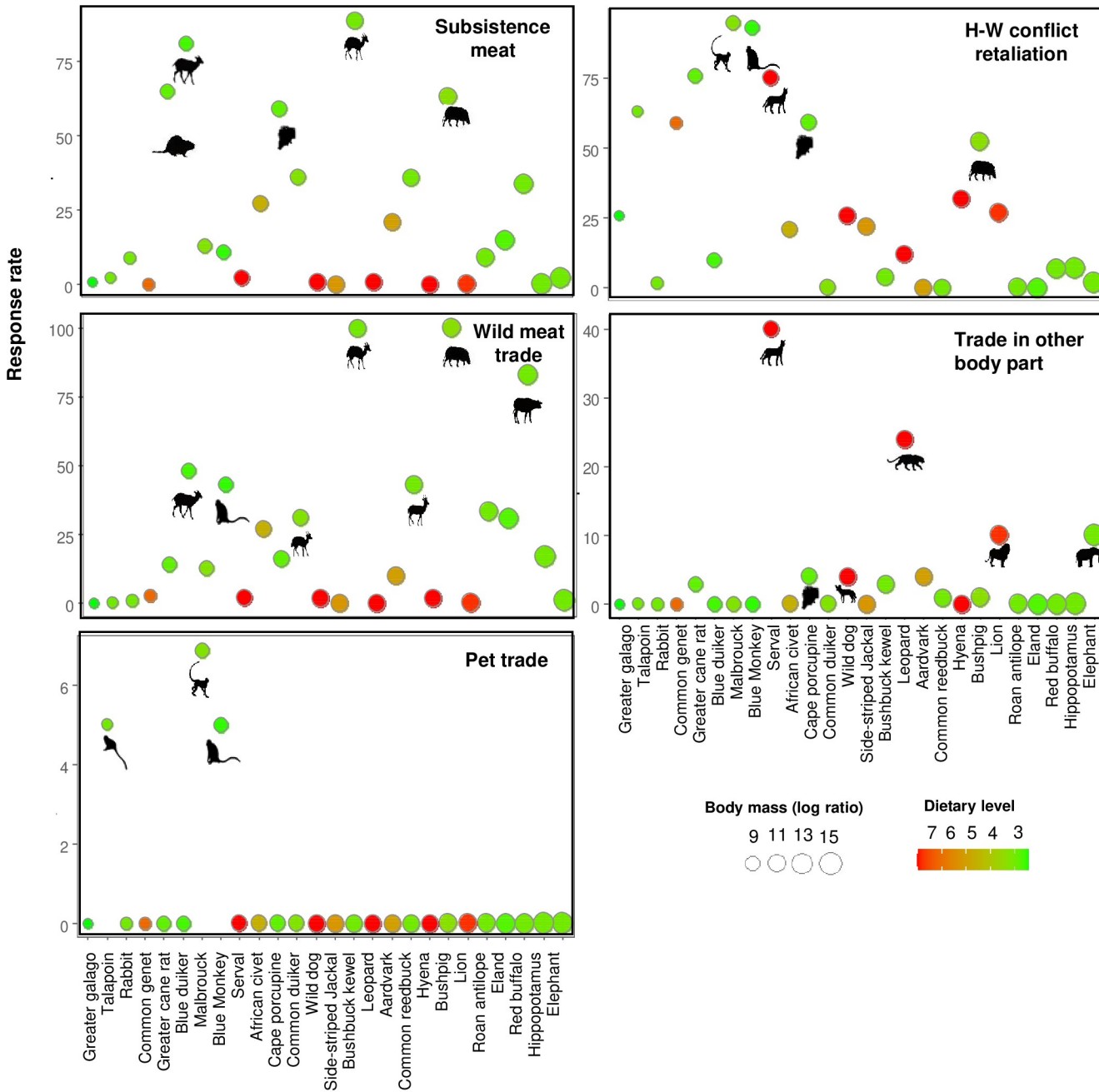

**Fig 4. Species selected to meet different types of hunting motivations.** Response rates were calculate from all interviewed hunters (n = 115). Species are denoted by their English names and ordered left to right from smallest to largest. Circle sizes are proportional to species body mass; circle colours denote trophic level in terms of dietary energy, ranging from green (lowest) to red (highest).

depleted, the size structure of prey species gradually shifted towards small to medium bodied species [14]. This explains why recent mammal captures are largely comprised of small to medium-sized species. In addition, even if the most important motivation associated with kills of large-bodied species was the wildmeat trade, medium and small-bodied species were also

**Table 1. Model selection and details of the models using generalized linear mixed models (GLMMs) to examine the effects of body size and dietary level on the number of hunters reporting each form of hunting motivation for killing any given species.**

| Response variable | Predictors | Estimate | Std. Error | z value | Pr(>|z|) | | AIC | ΔAIC |
|---|---|---|---|---|---|---|---|---|
| Subsistence meat | **Body mass** | **-0.00000174** | **6.96E-07** | **-2.502** | **0.0124** | * | **2604.5** | **0** |
| | **Trophic level** | **-0.763** | **5.18E-02** | **-14.76** | **<2e-16** | *** | | |
| | **Body mass** | **0.02763** | **0.08581** | **0.322** | **0.7475** | | **2606.2** | **1.7** |
| | **Trophic level** | **-0.64927** | **0.28684** | **-2.263** | **0.0236** | * | | |
| | **Body mass: trophic level** | **0.0157** | **0.02774** | **0.566** | **0.5714** | | | |
| | Trophic level | -0.48553 | 0.03699 | -13.13 | <2e-16 | *** | 2618.3 | 13.8 |
| | Bodymass | 0.07622 | 0.01962 | 3.886 | 0.0001 | *** | 2826.5 | 222 |
| | Mo | | | | | | 3117.1 | 512.6 |
| Wild meat trade | **Energetic level** | **-0.52743** | **0.05399** | **-9.768** | **<2e-16** | *** | **2844.1** | **0** |
| | **Body mass** | **4.49E-04** | **6.97E-04** | **-0.644** | **0.519** | | **2846** | **1.9** |
| | **Trophic level** | **-7.29E-01** | **4.93E-02** | **-14.79** | **<2e-16** | *** | | |
| | Body mass | 0.15831 | 0.11619 | 1.363 | 0.173 | | 2847 | 2.9 |
| | Trophic level | 0.02565 | 0.37307 | 0.069 | 0.945 | | | |
| | Body mass:Trophic level | -0.05582 | 0.03783 | -1.475 | 0.14 | | | |
| | log(bodymass) | 0.01854 | 0.02575 | -0.72 | 0.472 | | 2986 | 141.9 |
| | Mo | | | | | | 3260.7 | 416.6 |
| Trade in other body parts | **Body mass** | **2.94E-04** | **7.03E-04** | **0** | **0.675** | | **1006.6** | **0** |
| | **Trophic level** | **4.11E-02** | **5.11E+01** | **10.59** | **<2e-16** | *** | | |
| | **Body mass** | **-1.50E-01** | **1.53E-01** | **-0.979** | **0.3278** | | **1007.8** | **1.2** |
| | **Trophic level** | **5.51E-01** | **3.00E-01** | **1.835** | **0.06655** | . | | |
| | **Body mass: trophic level** | **2.59E-02** | **2.88E-02** | **0.899** | **0.36887** | | | |
| | Trophic level | 8.17E-01 | 5.08E-02 | 16.09 | <2e-16 | *** | 1044.8 | 38.2 |
| | Mo | | | | | | 1113.9 | 107.3 |
| | Body mass | 9.02E-04 | 3.18E-02 | 0.028 | 0.977 | | 1624.2 | 617.6 |
| Pet trade | **log(bodymass)** | **-8.90E-02** | **1.40E-03** | **-63.78** | **<2e-16** | *** | **376** | **0** |
| | **Trophic level** | **-1.257** | **4.18E-01** | **-3** | **0.0027** | ** | | |
| | **log(bodymass)** | **-1.92E+00** | **3.35E+00** | **0.572** | **0.057** | . | **377.1** | **1.097** |
| | **Trophic level** | **6.526** | **8.80E+00** | **0.741** | **0.466** | | | |
| | **log(bodymass): trophic level** | **-1.028** | **1.16E+00** | **-0.89** | **0.383** | | | |
| | Body mass | -1.4594 | 4.35E-01 | -3.355 | 0.00264 | ** | 386.46 | 10.463 |
| | Trophic level | -2.013 | 1.35E+00 | -1.492 | 0.0687 | | 398.64 | 22.641 |
| | Mo | | | | | | 479 | 103 |
| Retaliation | **Body mass** | **-1.74E-06** | **6.96E-07** | **-2.502** | **0.0124** | * | **2604.5** | **0** |
| | **Trophic level** | **-7.63E-01** | **5.18E-02** | **-14.76** | **<2e-16** | *** | | |
| | **log(bodymass)** | **0.188525** | **0.184788** | **1.02** | **0.308** | | **2606.3** | **1.8** |
| | **Trophic level** | **-0.456286** | **0.071617** | **-6.371** | **1.88E-10** | *** | | |
| | **Body mass: trophic level** | **-0.007241** | **0.018505** | **-0.391** | **0.696** | | | |
| | Trophic level | -0.45395 | 0.02545 | -17.83 | <2e-16 | *** | 2621.7 | 17.2 |
| | Body mass | 0.04466 | 0.02432 | 1.836 | 0.0663 | . | 3027.9 | 423.4 |
| | Mo | | | | | | 3602.4 | 997.9 |

Estimated values indicate the coefficients associated with the variable listed on the left. This represents the estimated amount by which the odds (that each response variable would increase if each explanatory variable were one unit higher). Z-values indicate the degree to which explanatory variables exert a significant effect. Pr (>|z) denote significance levels as following: ns P > 0.05

** P ≤ 0.01

*** P ≤ 0.001. AIC Akaike Information Criterion; ΔAIC difference of AIC with respect to the best model; Mo null model. Models with substantial support are in bold.

**Table 2. Mammal species and their respective body parts used for medicinal and "magical" purposes within Qui-çama National Park and Quiçama Game Reserve, Angola, Southwest Africa.**

| Species/Vernacular name | Ailment treated | Animal body part used | IUCN status |
|---|---|---|---|
| *Panthera leo*—Lion | Asthma | Fat | VU |
| *Panthera pardus*—Leopard | Ward off envy | Skin | VU |
| *Mellivora capensis*—Honey badger | Physical weakness | Bone | LC |
| *Tragelaphus scriptus*—Bushbuck kewel | Personal animosity | Skin | LC |

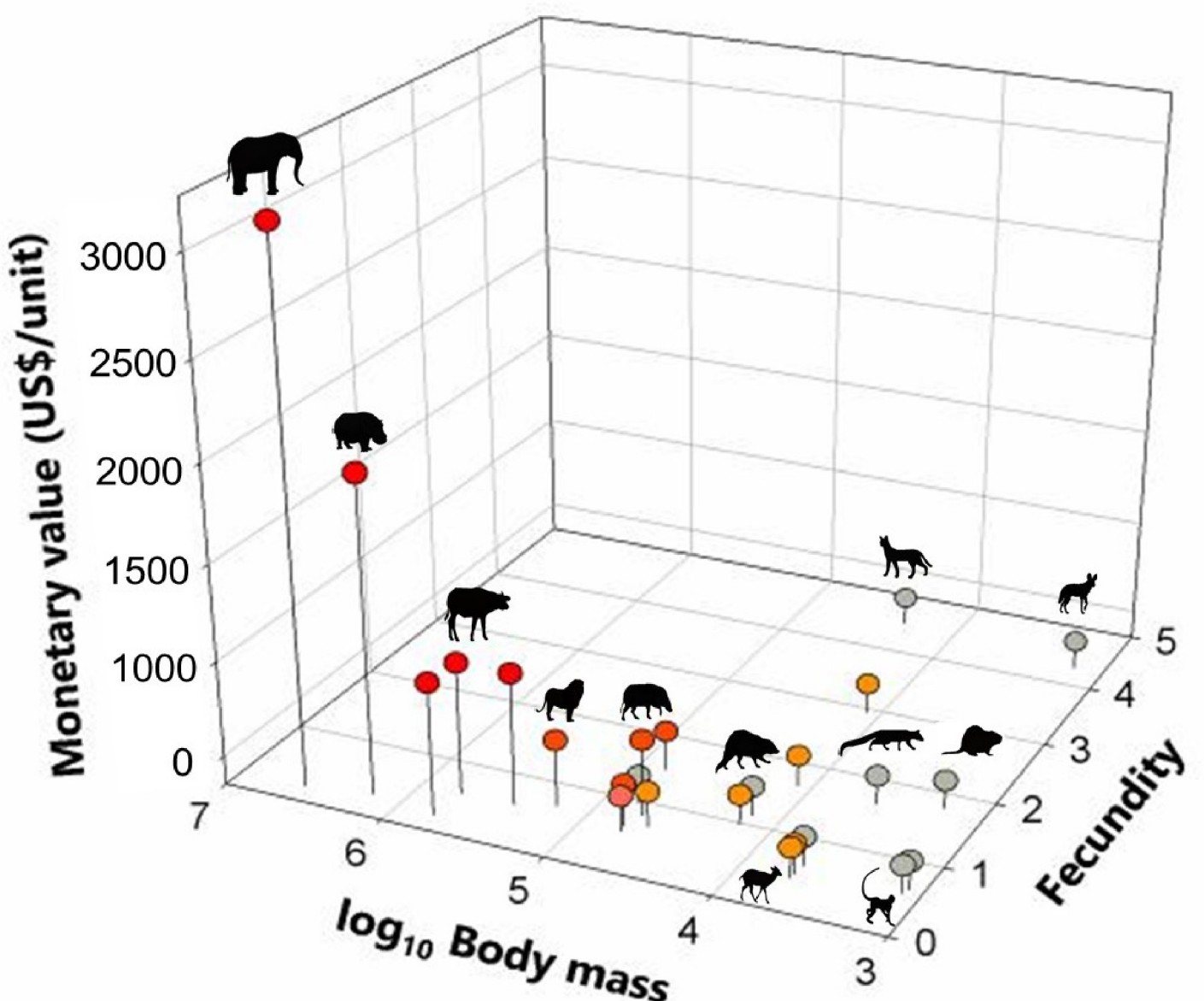

**Fig 5. Monetary values obtained from each prey item traded in the local market.** Mammal species are ordered along different axes i) z: From top to bottom from least to most profitable; ii) x: From left to right from largest to smallest; and iii) y: From left to right from lowest to highest fecundity rate. Warmer colours indicate higher offtakes in monetary values. Dollar values were based on the 2017 Kwanza conversion rate (US$1 = 165 Kwanzas).

captured to supply the bushmeat market. These species were also often captured for other reasons, such as subsistence needs and retaliation against wildlife posing threats.

Given the temptation to participate in the bushmeat trade, some informants explained that they were prepared to invest several days searching more remote areas of Quiçama in pursuit of large-bodied mammals even if their numbers had been depleted. This is consistent with a study in Indonesia, where market demand and supply of wild pigs was unaffected when local wild pig populations collapsed because hunters simply travelled farther in pursuit of game [54]. This hunter-prey system must, however, be carefully evaluated because despite the great effort by some hunters to capture species providing high energy and monetary returns, several species are absent from prey profiles in the recent years, which likely indicates severe population declines if not local extinctions. Among the five largest and most profitable species, red buffalos were recently slaughtered, but their remaining populations are now restricted to remote forest areas in the southern Quiçama region. Those buffalo populations, therefore, should receive immediate conservation attention, to prevent regional scale extirpation such as the case of native populations of Eland and Roan Antelope.

Intensification of hunting activities over the last decade can be understood given the dire post-war economic crisis in Angola, which is now further exacerbated by the COVID-19 outbreak that is plaguing the country. As a consequence, hunters from Northern Quiçama are hunting in the proximity or within the special protection zone of QNP, which corresponds to 10% of the park area that is patrolled on a daily basis by 15 rangers. This area and even the most remote portions of Quiçama have become increasingly accessible to not only local residents, but also outsiders in search of desirable game and other nontimber products. In a plight similar to that of red buffalos, native elephant populations at Quiçama are now restricted to forest landscapes where they are less detectable [14]. Elephant populations in Angola as a whole are experiencing intense poaching pressure as well as losing habitat to expanding human settlements. Based on observed carcass ratios, a study has shown that elephant populations in southern Angola started to increase after the war until 2005, but then began to decrease at an annual rate >2% 50). Although many landmines remain in southeast Angola following the civil war, and elephants have been killed by mines [55] poaching for ivory trade, rather than landmines, is the primary cause of mortality based on a large number of carcasses observed in a survey conducted in 2015 in southern Angola [56]. As a consequence, Angola is in the midst of one of the worst poaching crises of any country within the savannah range of African elephants and nationwide population declines will almost certainly continue if the drivers of mortality are not reversed.

Hunters reported that they often captured more than one individual larger than 40 kg each month to maximize income from bushmeat sales, but that those species often failed to contribute meat to their household subsistence for several months. In this case, only less desirable leftovers from dressed carcasses of large animals, such as the head and hooves, would be locally consumed, with consumption of whole prey items of large mammals in the hunter's household restricted to the most important holidays. This partially explains why body mass did not exert a significant effect on species selectivity in terms of community-level subsistence, given that more frequent local consumption of smaller-bodied species can be explained by both the higher abundance of those species and the fact that larger species were primarily directed to the bushmeat trade. However, another important factor to be considered is overall preference in terms of palatability. For example, interviewees reported that medium-sized mammals such as Blue Duiker (5 kg) have "soft tasting" meat compared to several larger species such as Red Buffalo (600 kg). Also, villagers typically complied with some important dietary taboos regarding primates, which were rarely consumed at Quiçama even though they were one of the most abundant mammal taxa larger than 1 kg.

Primates at Quiçama were killed primarily in retaliation against crop raiding and livestock depredation and to supply the pet trade, with exception of Blue Monkeys (6 kg), reported by many hunters as killed to fuel the bushmeat trade. Although primates were rarely used as a source of meat across the study area, this practice is common in other African and Neotropical sites [57], and they account for the largest number of mammal species threatened by hunting worldwide [5]. Given plausible scenarios of wildlife depletion, however, hunters in the future may become less selective and gradually change their hunting practices to include primates as game species, which could be directed to either the bushmeat trade or household subsistence [14]. Local use of mammals for medicinal purposes did not necessarily involve new kills since the animal parts used were essentially minor byproducts of prey acquisition to meet other purposes. In addition, there are cases in which the medicinal exploitation of wild mammals involved the non-invasive use of scats or dung, rather than actual mortality [31].

In Quiçama, the median abundance value of all large and medium-sized game species, and some small-bodied game species (Blue Duiker and Blue Monkey), in savannah areas declined during the war and did not recover during the post-war period. In contrast, in forest landscapes, 15.8% of all large and medium-sized game species, whose abundance had declined during the war, have since experienced post-war population recovery [14]. This helps explain why hunting for both for subsistence and meat trade encompassed a larger number of species in the northern savannahs than in the forest sites; since savannah areas are the most defaunated, prey acquisition in this environment follows a generalist hunting strategy that goes well beyond selection of preferred species. On the other hand, hunters in the southern savannah areas had access to alternative fish protein, which was either consumed locally or sold. Hunters at these sites would thus also go fishing but concentrate on the most desirable game species when hunting. In terms of non-food motivations, the number of reports of species killed for trade in other animal body parts and retaliation was higher in forest than in savannah areas, because forest sites hosted the highest abundance of species whose body parts are traded (such as leopard skin and elephant ivory); and crops and domestic livestock were nearer a wider spectrum of crop-raiders and larger carnivores. In recent years, residents of forest sites in the Center-South of Quiçama have asked the local government for support because elephants that emigrated from the region during the war years, are currently returning to their habitat and raiding local crops.

We found that species at the lowest dietary trophic levels were the most frequently captured for both the wild meat trade and local subsistence. This is consistent with data from kill profiles in Afrotropical forests showing that bushmeat harvest rates of frugivore–herbivores are the highest, both in terms of the number of carcasses (44.5%) and overall biomass (57.2%) [58]. However, carnivores were frequently involved in human-wildlife conflicts and were also affected by commercial use of body parts for both medicinal (bones and fat) and ornamental (skin and teeth) purposes. Many leopards were incidentally caught in traps set for antelopes, and then subsequently killed for the illegal wildlife trade. Given that carnivore skin prices are based on quality which is reduced by bullet perforations, some hunters at Quiçama either hanged the animal or waited for it to die from bleeding, rather than immediately killing the animal. Protecting hide quality from projectile damage also explains why many carnivores were poisoned to death rather than shot [59].

Considering the hunting motivations described here (subsistence, animal trade and conflict retaliation), similar motivations were found in a study carried out in Tanzania, which showed that food acquisition and generation of income (mentioned by 79% and 78% of all interviewees, respectively) were most frequently cited as to why hunters poached [27]. A review on hunting motivations in Vietnam also showed that crop protection and cash income were the main motivations behind hunting [60]. On the other hand, a survey conducted in Oregon,

USA, showed that recreational and trophy hunting were the main motivations driving deer hunting [61]. These differences can be partly explained by the low-income populations of villagers at Quiçama who are unlikely to go hunting unless this can generate either direct or indirect income. Our results also show positive hunter selectivity targeting adults, which we attributed to their larger size and greater profits from sales. Likewise, a global survey revealed that humans kill adult males and females, the reproductive capital of prey populations, up to 14 times more often than natural terrestrial predators, such as wolves or lions [2]. As a consequence, life-history traits may often shift to early reproduction since older animals are consistently eliminated by hunters [62]. We found that this also applied to males, which were generally larger and provided higher returns than females. In addition, interviewees indicated that males were more frequently captured because they are more "distracted" and consequently failed to detect active traps. Some studies have shown that such selection for adults of a single sex may lead higher reproductive investments [63,64]. Additionally, due to the logic dictating that larger animals are more frequently selected in both subsistence and commercial exploitation systems [20], phenotypic changes of populations selected by humans show a 20% decline in traits related to size and 25% in traits associated with reproductive stage. These values are higher than those observed in societies in which hunters select prey only on the basis of local consumption [64].

In relation to division of labour and social norms, there was a male-bias in hunters involved in the bushmeat trade, whereas women primarily pursued smaller-bodied species for either local consumption or protecting livestock. Among modern hunter-gatherers, female exclusion from hunting large mammals is apparently closely related to the manufacture and use of traditional or fire weapons, and associated economic and/or religious norms [65]. Women often have no hunting weapons of their own so their hunting activities under those restrictions are confined to small-bodied prey. A number of anthropologists agree that the division of labour among such groups stems from culturally derived, gender-based beliefs, the importance attributed to big-game hunting, and the convertibility of these into prestige and authority that preclude female access to essential weapons, rather than fundamental sex-based anatomical or physiological differences. Male-biased hunting effort allocates to men society's most valued labour and opportunities to distribute high-value foods. Division of labour in relation to prey acquisition is thus grounded in gender politics rather than inherent sex differences [65].

Regarding the motivations across different sites, the roads that traverse savannah habitats in Northern to Southern Quiçama serves as an important route for the wildlife traffic. The greater difficulty in accessing and navigating forest areas has rendered hunting activities therein less intensive than in savannah areas [14], but we found that established traffic routes have connected cities like Muxima and Dondo to forest refugia serving as the last stronghold for buffalo and large carnivore populations. Another threat is the marked growth of urban development and extractive activities in Quiçama coastal areas. The detrimental effects of human incursion into Angolan protected areas on the abundance and distribution of large mammal species underscore the need for intensive mitigation of anthropogenic threats to restore the high wildlife conservation value in all areas impacted by the civil war [66]. Also, the problem is collective, given that hunting is carried out by both residents and outsiders, often motivated by urban commerce and illegal traders of ammunition and fire weapons. Some of the arsenal typically used by hunters at Quiçama consisted of new ammunition and automatic rifles, which indicate that they were sourced from military bases within the park, which were installed during the Angolan civil war.

Our results show that hunting is considerably more profitable in the short term compared to alternative sources of local income. It also pays to go hunting because enforcement against

(and penalties related to) poaching are virtually non-existent and people cannot, and will not, protect species and natural habitats as a long-term strategy if their short-term needs are not met [27,67]. However, these benefits are modest, largely unsustainable and come at a high price. They are modest because local hunters receive only the smallest share of the profits from the illegal wildlife supply chain, yet this follows a logic rationale because alternative opportunities to derive benefits from wildlife are unavailable. For example, although the income from a Cape Bushbuck sale was on average US$41 (24000kz) for a local hunter, the profits associated with selling this animal to an urban restaurant could reach US$100 (36000kz). The price discrepancy of ivory per kilogram was even greater, with local hunters earning less than US$50/ kg of ivory, whereas this can fetch US$2150/kg in international markets despite the fact that ivory prices have plummeted since 2014 [68]. Needless to say, the ivory trade is largely unsustainable and comes at a high price in several African countries, with the value generated from the ivory trade corresponding to only 20% (US$85/km$^2$) of the value generated from ecotourism [69,70]. A loss of up to US$41.1 million in potential annual income has therefore been estimated for some countries as consequence of bushmeat hunting. Hence, poaching should not be seen as a solution to the lack of alternative livelihoods because its food security benefits are generally unsustainable and often brings fewer economic and livelihood benefits compared to ecotourism and trophy hunting [69,70].

The traditional population of Quiçama remains disenfranchised, in terms of healthcare, education, income opportunities and other rights. The lack of institutional capacity to effectively mitigate anthropogenic threats to both biodiversity and traditional communities at Quiçama is often justified by many as an ongoing consequence of the civil war, but there are examples that wildlife populations can recover through protection, and local residents can be ethically assisted even in the aftermath of the war. For example, after peace was declared, the Mozambique government partnered with a Foundation to restore Gorongosa National Park. Those living around parks received assistance for farming, education and healthcare and several hundreds also found jobs in the park [71]. A persistent absence of institutional enforcement in countering threats to wildlife could thus be a lingering effect of sustained armed conflicts, but it should not always be justified as an obstacle to nature conservation.

Although the mammal fauna of Quiçama is exposed to mounting perils, such as human population growth, deforestation and spread of exotic species, bushmeat exploitation for commerce is undoubtedly the most imperative threat. We therefore argue for the imminent implementation of activities that can ensure both wildlife recovery, and food security and egalitarian rights for the local human communities. These efforts should include environmental education, greater human rights and social equality, and the understanding and integration of legal anti-poaching instruments and hunting management. Given the frequent engagement of residents with hunting and the highly scenic landscape value of the study area, we strongly encourage community-based ecotourism and wildlife management activities. In addition, considering that local ecological knowledge can efficiently strengthen management and conservation guidelines, we also highlight the importance of employing local villagers as wildlife guides and including them in official wildlife management and monitoring plans. We also draw attention to the need to implement adequate anti-poaching enforcement and full-time presence of park staff in the central and southern portions of Quiçama, especially in areas exposed to the most intensive hunting activity and along roads through which the wildlife trade operates. Finally, the routes that link Quiçama to large cities must be monitored and transformed from traffic routes into highways that enable sustainable activities that capitalize on the regional ecological and cultural diversity, such as educational endeavors for school kids and ecotourism.

## Supporting information

**S1 Fig. Frequency that each hunting motivation was reported for each species.** Report rates were calculate from the number of hunters reporting capturing each species for each motivation. Species are denoted by their English names and ordered left to right in accordance to the frequency of meat trade as a motivation for capturing each species.
(PDF)

**S2 Fig. Species compositional similarity in terms of hunting motivations.** It was obtain from (a) a multiple factor analysis (MFA). Contributions of each hunting motivation and interviewee sex and age for (b) the first ($Dim_1$) and (c) the second dimension ($Dim_2$). The red dotted line indicates the percentage that would be obtained if all factors contributed equally to the overall variance.
(PDF)

**S3 Fig. Species compositional similarity of each site (structured according to landscape structure and geographic position) in terms of hunting motivations obtained from a multiple factor analysis (MFA).**
(PDF)

**S1 Table. Nonvolant terrestrial game mammal species larger than 1 kg recorded in hunting motivation within Quiçama National Park and Quiçama Game Reserve, Angola.**
(PDF)

**S1 File. Questionnaire (English version) and Questionário (versão em português).**
(PDF)

## Acknowledgments

We sincerely thank the people from local communities who shared their knowledge on wildlife and supported us in data collection through opportunist observations. Without them, this work would not be possible. We would like to thank the Environmental Ministry of Angola, Quiçama municipal administration, Quiçama Park administration and all village coordinators for authorizing and support this research. We thank the reviewer and the scientific Editor for the important contribution to this manuscript.

## Author Contributions

**Conceptualization:** Franciany Braga-Pereira, Rômulo Romeu da Nóbrega Alves, Carmén Van-Dúnem Santos.

**Data curation:** Franciany Braga-Pereira.

**Formal analysis:** Franciany Braga-Pereira, Carlos A. Peres.

**Funding acquisition:** Franciany Braga-Pereira, Carmén Van-Dúnem Santos.

**Investigation:** Franciany Braga-Pereira.

**Methodology:** Franciany Braga-Pereira.

**Project administration:** Franciany Braga-Pereira.

**Supervision:** Rômulo Romeu da Nóbrega Alves.

**Writing – original draft:** Franciany Braga-Pereira.

**Writing – review & editing:** Franciany Braga-Pereira, Carlos A. Peres, Rômulo Romeu da Nóbrega Alves, Carmén Van-Dúnem Santos.

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
