## [Decision Letter · Decision Letter 0]

11 Jun 2021

PONE-D-21-09230

Intrinsic and extrinsic motivations governing prey choice by hunters in a post-war African forest-savannah macromosaic

PLOS ONE

Dear Dr. Braga-Pereira,

Thank you for submitting your manuscript to PLOS ONE. After careful consideration, we feel that it has merit but does not fully meet PLOS ONE’s publication criteria as it currently stands. Therefore, we invite you to submit a revised version of the manuscript that addresses the points raised during the review process.

Thank you for submitting your manuscript to PLOS ONE. After careful consideration, we feel that it has merit but does not fully meet PLOS ONE’s publication criteria as it currently stands. Therefore, we invite you to submit a revised version of the manuscript that addresses the points raised during the review process.

As you will see, our referee underlined the interest of considering the interviewed occupation in the motivations for prey acquisition objective. Other interesting methodological details have also been underlined in the review report, please consider all of them in a further version of you work.

On the other hand, test of null hypotheses and AIC model selection should not be used together. In fact, your AIC model selection will allow you to select the most parsimonious model including significant and non-significant variables (see page 83 of Burnham and Anderson.2002 Model selection and multimodel inference book). Them, the effect size of specific variables should be assessed by their relative importance or the R2. P-values are not a measure of their importance. (see Halsey Lewis G. 2019. The reign of the p-value is over: what alternative analyses could we employ to fill the power vacuum? Biol. Lett.152019017420190174). In any case, if you decide to use p-values, please provide the full statistics (F-value, df, p-value) instead of naked p-values.

We look forward to receiving your revised manuscript.

Kind regards,

Emmanuel Serrano, PhD

Academic Editor

PLOS ONE

Journal Requirements:

4. We noted in your submission details that a portion of your manuscript may have been presented or published elsewhere. [Only the map in this manuscript is also in other two articles, submeted as a related manuscript file] Please clarify whether this publication was peer-reviewed and formally published. If this work was previously peer-reviewed and published, in the cover letter please provide the reason that this work does not constitute dual publication and should be included in the current manuscript.

6. We note that Figure 1 in your submission contain map images which may be copyrighted. All PLOS content is published under the Creative Commons Attribution License (CC BY 4.0), which means that the manuscript, images, and Supporting Information files will be freely available online, and any third party is permitted to access, download, copy, distribute, and use these materials in any way, even commercially, with proper attribution. For these reasons, we cannot publish previously copyrighted maps or satellite images created using proprietary data, such as Google software (Google Maps, Street View, and Earth). For more information, see our copyright guidelines: http://journals.plos.org/plosone/s/licenses-and-copyright.

6.1.    You may seek permission from the original copyright holder of Figure 1 to publish the content specifically under the CC BY 4.0 license. 

6.2.    If you are unable to obtain permission from the original copyright holder to publish these figures under the CC BY 4.0 license or if the copyright holder’s requirements are incompatible with the CC BY 4.0 license, please either i) remove the figure or ii) supply a replacement figure that complies with the CC BY 4.0 license. Please check copyright information on all replacement figures and update the figure caption with source information. If applicable, please specify in the figure caption text when a figure is similar but not identical to the original image and is therefore for illustrative purposes only.

Reviewers' comments:

Reviewer's Responses to Questions

**Comments to the Author**

1. Is the manuscript technically sound, and do the data support the conclusions?

Reviewer #1: Yes

2. Has the statistical analysis been performed appropriately and rigorously? 

Reviewer #1: Yes

3. Have the authors made all data underlying the findings in their manuscript fully available?

Reviewer #1: Yes

4. Is the manuscript presented in an intelligible fashion and written in standard English?

Reviewer #1: Yes

5. Review Comments to the Author

Reviewer #1: The main merits of this manuscript are (1) using local knowledge to provide an interesting, useful and easy methodology to (2) estimate the motivations of hunters in a post-war area. In many places worldwide, the only data available for management and conservation proposes is the ecological local knowledge (besides hunting bags when available), therefore any method focused on using this knowledge to produce population estimates (management) and the motivations for killing certain species (conservation) will be proven very helpful, especially when referring to natural areas or countries with strong budget limitations or low-income economies.

As a matter of fact, it is impossible to change a person's behavior towards wildlife (e.g., feeding or poaching) without knowing the motivations behind that behavior. This study addresses a stated need of new strategies which, based on the motivations towards hunting certain species, help to better plan for the management of ecosystems that are recovering from strong disturbances, such as wars, and where poaching is present with strong impact on the recovery of wildlife populations.

This is a sound-based manuscript; however, I have some remarks on the text

General comments:

L79-82: I may not have understood this part of the manuscript correctly, so in that case the authors should state it clearer, but when asking about the motivation behind the pursuit each wild mammals during the semi-structured individual interviews, did the authors used a closed questionary based on Muth and Bowe 1998, and the rest of the intrinsic, extrinsic and individual based motivations mentioned in the manuscript, or did you allowed the hunters to relate their motivations without any “constrain”?

I ask this question since you the authors had the opportunity to collect first-hand the motivations of hunters in a post-war area, I think that the appearance of unpredicted motivations could have had a great importance, both for the results and for the impact of the present manuscript.

Moreover, I think the discussion of the manuscript could be improved if the results of this study were compared with other studies, stating similarities and differences with the motivations of poachers in other post-war areas (if any, and ideally in other continents), or in non-post-war areas, such as the study cited Muth and Bowe 1998, in the USA, where the conditions for poachers were totally different.

L211 Data analysis.

I have several concerns about the data analysis:

The authors interviewed 115 hunters with different ages (19-80 years old), from eight settlements (one located in the Game reserve), with different population densities and located in different ecosystems (forest-savannah). If I was a manager of this or a similar natural area wanting to implement conservation strategies based on working with the communities, I would find attractive your study but after reading the manuscript, several questions would come to my mind. Are the motivations for hunting the same along all ages and settlements regardless of their differences? Besides stating the differences in the activity among male and female and adults, children and the elderly, the manuscript would be improved if the authors evaluated possible differences in the motivations due to the hunters’ characteristics.

As the authors did not evaluated differences due to hunters’ characteristics, why did the authors used a GLM instead of a GLMM, where the hunter and the settlement were included as nested random effects? The results would be more robust.

But as I stated before, I think that the authors are not taking full advantage of what they could get from the data. In my opinion and based on my experience, to study the motivations of human behaviors towards wildlife based on surveys, the multivariate statistical tests can give you more information, especially those based on the canonical ordination, such as multiple factor analysis (http://www.sthda.com/english/articles/31-principal-component-methods-in-r-practical-guide/116-mfa-multiple-factor-analysis-in-r-essentials/).

Minor comments:

L44-46: I recommend to review the citations in text and references. Just as an example I found two mistakes between L82 and L86 (e.g. In L82 you cited (Muth and Bowe 1998) but I cannot find the citation in references, L86-87 you cited as (Knapp and Peace 2017) but the article has three authors.

L199-201: Again, I may not have understood this part correctly, if so, the authors should state it clearer. In the example: “if the grazer/frugivore Tragelaphus oryx consumes 70% grass and 30% fruit, its mean trophic energy level would be 2.7 (= (0.7 × 3) + (0.3 × 2)”, but according to the levels in L195-197 ((1) grazer/folivore < (2) frugivore < (3) granivore…) the level should be 1.3 (= (0.7 × 1) + (0.3 × 2).

L343-367: The authors should state the usefulness of ecological local knowledge as a tool to estimate wildlife abundance.

6. PLOS authors have the option to publish the peer review history of their article (what does this mean?). If published, this will include your full peer review and any attached files.

Reviewer #1: No

---

## [Author Response · Author response to Decision Letter 0]

19 Nov 2021

To: Emmanuel Serrano

Academic Editor of Plos One

Dear Editor,

We are submitting a new version of our manuscript (ID: PONE-D-21-09230)

entitled “Intrinsic and extrinsic motivations governing prey choice by

hunters in a post-war African forest-savannah macromosaic”. We would

like to thank the editor and reviewer for the comprehensive comments and suggestions on

the paper. We appreciate the feedback and believe that responding to these

comments have improved the quality of our manuscript.

Please find below a point-by-point response to the comments provided by the Editor and

Reviewer. We are providing two versions of the Manuscript, one with and other

without track changes, so that you can see exactly what has been altered. Note

that the lines indicated below is referring to the clean version of the Manuscript.

Yours sincerely

Franciany G. Braga-Pereira (on behalf of all co-authors)Journal: Plos One

Ref:PONE-D-21-09230

We included in the analysis results of the linear models the F-value, df and

Adjusted R-squared (lines 252-257; 370-373). For generalized linear models and

generalized linear mixed models we used residual checks to verify whether our

models were, in principle, suitable or otherwise. Finally, we used the Akaike

information criterion to select models of interest if ∆AIC values >6 (∆AIC obtain

from the difference between a null and complete model AIC values). ΔAIC>6 was

chosen following Harrison et al., (2018) and Richards (2008) (lines 240-243; 265-

268).

We included the questionnaire in both in the original language (Portuguese) and

in English, as Supporting Information.

The map we published in Elsevier

(https://www.sciencedirect.com/science/article/abs/pii/S0006320720308028) is

under copyright, which means the map used in the previous version of this article

was also under copyright. We contacted Elsevier to request a license to use the

images, however we didn’t get a return, so we made a new map.

Because of that, below, we have included two map options. Option 1 is similar

the original map, but not the same. If it is possible to use this option 1, it would

interest as it demarcates the savannah and forest landscapes. If, because of

the similarity between this new map and the original one, you choose a more

different image; we have created a second map option (Option 2)

People involved in the research answered questions about hunting, which is an

illegal activity in the region. For this reason, we need to maintain the anonymityof informants, because data contain potentially sensitive information imposed

by UFPB ethics committee (email: comitedeetica@ccs.ufpb.br).

Results presented herein are entirely original and this paper is not being

considered for publication in other journal. All sources of funding are

acknowledged in the manuscript, and all appropriate ethics and other approvals

were obtained for this research. This research project was registered in the

Plataforma Brasil, which is a Brazilian database of research records involving

human communities and prior research approval was obtained from the Research

Ethics Committee of the Health Sciences Center at the Federal University of

Paraiba - (CEP -CCS- UFPB-CAAE 59846816.3.0000.5188). This study was also

authorized by the Environmental Ministry of Angola (Ministério do Ambiente de

Angola) based on license number 148INBAC. MINAMB/2016 and by all local

community leaders of human settlements in which this research was conducted.

We also received a municipal authorization from the mayor of Quiçama (city of

Angola) via license number 017 / GAB.ADM.M.Q / 2017.

We confirm the first author of the manuscript submitted to Plos One (Franciany

Braga-Pereira) is the author of the map used in the manuscript.

Reviewer #1: The main merits of this manuscript are (1) using local knowledge

to provide an interesting, useful and easy methodology to (2) estimate the

motivations of hunters in a post-war area. In many places worldwide, the only

data available for management and conservation proposes is the ecological local

knowledge (besides hunting bags when available), therefore any method focused

on using this knowledge to produce population estimates (management) and the

motivations for killing certain species (conservation) will be proven very helpful,

especially when referring to natural areas or countries with strong budget

limitations or low-income economies.

As a matter of fact, it is impossible to change a person's behavior towards wildlife

(e.g., feeding or poaching) without knowing the motivations behind that

behaviour. This study addresses a stated need of new strategies which, based on

the motivations towards hunting certain species, help to better plan for the

management of ecosystems that are recovering from strong disturbances, such

as wars, and where poaching is present with strong impact on the recovery of

wildlife populations.

This is a sound-based manuscript; however, I have some remarks on the text.

Authors: We are very grateful for these positive comments and effort to improve

our manuscript. We followed all of your suggestions and we believe that our

manuscript has been further strengthened after these recommendations.

Reviewer #1

L79-82: I may not have understood this part of the manuscript correctly, so in

that case the authors should state it clearer, but when asking about the

motivation behind the pursuit each wild mammals during the semi-structured

individual interviews, did the authors used a closed questionary based on Muth

and Bowe 1998, and the rest of the intrinsic, extrinsic and individual basedmotivations mentioned in the manuscript, or did you allowed the hunters to relate

their motivations without any “constrain”?

I ask this question since you the authors had the opportunity to collect first-hand

the motivations of hunters in a post-war area, I think that the appearance of

unpredicted motivations could have had a great importance, both for the results

and for the impact of the present manuscript.

Authors: We agree with your comments regarding the opportunity to first-hand

collect the motivations of hunters and we confirmed we allowed the hunters to

relate their motivations without any “constrain”. We had improve the main

manuscript for a better explanation on that (lines 186-199) and we provided the

whole questionnaire used in this study in the supplementary material.

Reviewer #1 Moreover, I think the discussion of the manuscript could be

improved if the results of this study were compared with other studies, stating

similarities and differences with the motivations of poachers in other post-war

areas (if any, and ideally in other continents), or in non-post-war areas, such as

the study cited Muth and Bowe 1998, in the USA, where the conditions for

poachers were totally different.

Authors: We agree with you and had improved the discussion on that (lines 452-

460; 510-514; 521-545)

Reviewer 1# L211 Data analysis. I have several concerns about the data

analysis: The authors interviewed 115 hunters with different ages (19-80 years

old), from eight settlements (one located in the Game reserve), with different

population densities and located in different ecosystems (forest-savannah). If I

was a manager of this or a similar natural area wanting to implement

conservation strategies based on working with the communities, I would find

attractive your study but after reading the manuscript, several questions would

come to my mind. Are the motivations for hunting the same along all ages and

settlements regardless of their differences? Besides stating the differences in theactivity among male and female and adults, children and the elderly, the

manuscript would be improved if the authors evaluated possible differences in

the motivations due to the hunters’ characteristics.

As the authors did not evaluated differences due to hunters’ characteristics, why

did the authors used a GLM instead of a GLMM, where the hunter and the

settlement were included as nested random effects? The results would be more

robust.

But as I stated before, I think that the authors are not taking full advantage of

what they could get from the data. In my opinion and based on my experience,

to study the motivations of human behaviors towards wildlife based on surveys,

the multivariate statistical tests can give you more information, especially those

based on the canonical ordination, such as multiple factor analysis

(http://www.sthda.com/english/articles/31-principal-component-methods-in-rpractical-guide/116-mfa-multiple-factor-analysis-in-r-essentials/).

Authors: We appreciated this comment. We had performed a MFA for better

understand the similarities of species selected to attend different hunting

motivations regarding species body mass and energetic level; sampled site; and

interviewee age and sex.

In addition, to examine the effects of species body mass and trophic level; and

interviewee age, gender and natal community on the number of hunters reporting

each hunting motivation for killing any given species we performed generalized

linear mixed models (GLMM) and generalized linear models (GLM). We performed

a GLMM considering the interviewee as a random variable; and the body mass

and energetic level as fixed predict variable.

We assigned the 8 sampled communities into four study sites according to type

of landscape structure (forest of savannah) and geographic position. We

therefore sampled two communities in savannah areas of northern Quiçama; two

in savannah areas of southern Quiçama; two in central forest areas; and two

communities in southern forest areas. By this way we could better explain our

results for readers not familiar to the area and in the same time to ensure the

interviewee confidentiality, as some communities contained few hunters.See lines in the Methods section: 217-228; 232-249; 260-272.

See lines in the Results section: 313-328; Table 1 (355); 396-409.

See lines in the Discussion section: 489-509; 562-567.

See Supplementary Material: Figures 2 and 3.

Reviewer 1#L44-46: I recommend to review the citations in text and

references. Just as an example I found two mistakes between L82 and L86 (e.g.

In L82 you cited (Muth and Bowe 1998) but I cannot find the citation in

references, L86-87 you cited as (Knapp and Peace 2017) but the article has three

authors.

Authors: Thank you for your comment. We have reviewed the citations and

references.

Reviewer1# L199-201: Again, I may not have understood this part correctly, if

so, the authors should state it clearer. In the example: “if the grazer/frugivore

Tragelaphus oryx consumes 70% grass and 30% fruit, its mean trophic energy

level would be 2.7 (= (0.7 × 3) + (0.3 × 2)”, but according to the levels in L195-

197 ((1) grazer/folivore < (2) frugivore < (3) granivore…) the level should be 1.3

(= (0.7 × 1) + (0.3 × 2).

Authors: Thank you for your comment. We corrected the text (line 203-208).

Reviewer 1#L343-367: The authors should state the usefulness of ecological

local knowledge as a tool to estimate wildlife abundance.

Authors:

Thank you for your comment. We agree and had edited the text (lines:621-624)

---

## [Editor Report · Decision Letter 1]

22 Nov 2021

PONE-D-21-09230R1Intrinsic and extrinsic motivations governing prey choice by hunters in a post-war African forest-savannah macromosaicPLOS ONE

Dear Dr. Braga-Pereira,

Thank you for submitting your manuscript to PLOS ONE. After careful consideration, we feel that it has merit but does not fully meet PLOS ONE’s publication criteria as it currently stands. Therefore, we invite you to submit a revised version of the manuscript that addresses the points raised during the review process.

I am so sorry for the delays in the review of this interesting work. We have just to deal with some minor comments (about including some paraph about the use of questionaries and others about how did you perform model selection based on AIC values) before the final acceptance of your work.

I look forward to hearing from your revised work

Sincerely yours

Emmanuel

We look forward to receiving your revised manuscript.

Kind regards,

Emmanuel Serrano, PhD

Academic Editor

PLOS ONE
---

## [Author Response · Author response to Decision Letter 1]

25 Nov 2021

We included a sentence on the use of interviews in the introduction section and also included some references (lines 113-119).

We decided to follow the paper you suggested us and had included all possible models and compered their AIC values (247-250, 353-362). The model with the lowest AIC was retained, and the remaining competing models were ordered according to their Akaike differences (ΔAIC) with respect to the best model (lowest AIC) following Burnham & Anderson (2003).

---

## [Editor Report · Decision Letter 2]

26 Nov 2021

Intrinsic and extrinsic motivations governing prey choice by hunters in a post-war African forest-savannah macromosaic

PONE-D-21-09230R2

Dear Dr. Braga-Pereira,

We’re pleased to inform you that your manuscript has been judged scientifically suitable for publication and will be formally accepted for publication once it meets all outstanding technical requirements.

Kind regards,

Emmanuel Serrano, PhD

Academic Editor

PLOS ONE
---

## [Editor Report · Acceptance letter]

7 Dec 2021

PONE-D-21-09230R2 

Intrinsic and extrinsic motivations governing prey choice by hunters in a post-war African forest-savannah macromosaic 

Dear Dr. Braga-Pereira:

I'm pleased to inform you that your manuscript has been deemed suitable for publication in PLOS ONE. Congratulations! Your manuscript is now with our production department. 

Kind regards, 

on behalf of

Dr. Emmanuel Serrano 

Academic Editor

PLOS ONE